# Evaluation of Serum FGF21 Levels in Patients with Mitochondrial Aminoacyl-tRNA Synthetase Deficiency

**DOI:** 10.3390/ijms26199525

**Published:** 2025-09-29

**Authors:** Sebnem Tekin Neijmann, Dilek Gunes, Meryem Karaca, Volkan Karaman, Mehmet Cihan Balci, Gulden Fatma Gokcay, Asuman Gedikbasi

**Affiliations:** 1Department of Interdisciplinary Rare Diseases, Institute of Health Sciences, Istanbul University, 34452 Istanbul, Turkey; sebnemtekin@gmail.com; 2Department of Pediatrics, Division of Pediatric Nutrition and Metabolism, Istanbul Faculty of Medicine, Istanbul University, 34452 Istanbul, Turkey; dilek.gunes@istanbul.edu.tr (D.G.); meryem.karaca@istanbul.edu.tr (M.K.); mcbalci@istanbul.edu.tr (M.C.B.); ghuner@istanbul.edu.tr (G.F.G.); 3Department of Medical Genetics, Istanbul Faculty of Medicine, Istanbul University, 34452 Istanbul, Turkey; volkan.karaman@istanbul.edu.tr; 4Department of Pediatric Basic Sciences, Institute of Child Health, Istanbul University, 34452 Istanbul, Turkey

**Keywords:** mitochondrial disease, mitochondrial aminoacyl tRNA synthetase deficiency, FGF21, *AARS2*, *EARS2*, *DARS2*, *SARS2*, *WARS2*

## Abstract

Fibroblast growth factor 21 (FGF21), a pleiotropic hormone, is a significant modulator of energy homeostasis. We evaluated serum FGF21 levels in patients with a deficiency of mitochondrial aminoacyl-tRNA synthetase (mt-aARSs). Six patients with mitochondrial aminoacyl tRNA synthetase deficiency and twelve healthy volunteers were included in this study. Whole-exome sequencing was used for molecular diagnosis. Serum FGF21 levels in the case group and healthy volunteers were analyzed using the enzyme-linked immunosorbent assay. Exome sequencing test revealed nine different pathogenic variants in the *AARS2*, *EARS2*, *DARS2*, *SARS2*, and *WARS2* genes. A statistically significant difference was found between the serum FGF21 levels of the case and control groups: case group (n = 6), 882.49 ± 923.60 pg/mL; control group (n = 12), 20.89 ± 2.63 pg/mL (*p* < 0.001). The area under the ROC curve for FGF21 in the differential diagnosis of mitochondrial aminoacyl-tRNA synthetase deficiency was 1.000 (0.813–1.000). Sensitivity and specificity were 100%, and positive and negative predictive values were also 100% for an FGF21 cut-off value > 27.4 pg/mL. Assessment of FGF 21 levels as an indicator of mitochondrial damage in mt-aARSs deficiency may provide insight into the level of damage. Investigation of the biochemical mechanisms underlying the different levels of damage caused by different aminoacyl tRNA synthetases will be important in terms of elucidating clinical heterogeneity.

## 1. Introduction

The energy-producing organelles in cells, known as mitochondria, can be affected by a variety of physiological and genetic conditions in the body [1]. Mitochondria form dynamic networks that interact with other organelles, the cytoplasm, and each other to exchange chemicals and information. It is increasingly recognized that communication between mitochondria can cross cellular boundaries and even distinct tissues. Many common conditions, including metabolic disorders, neurological diseases, and cancer, are associated with mitochondrial stress and the dysregulated mitochondrial unfolded protein response (UPRmt). Mitokines are essential inter-organ communication mediators that impact systemic metabolic and physiological processes. They are signaling molecules that are released by the mitochondrial stress response and UPRmt [2]. Genetic and environmental factors leading to mitochondrial dysfunction can trigger a broader cellular stress response that releases mitokines such as fibroblast growth factor 21 (FGF21).

Mitochondrial diseases (MDs), caused by mutations in the nuclear and mitochondrial genomes, constitute an important group of rare inherited metabolic disorders. Rare diseases have been emphasized in recent years as they shed light on the pathophysiology of common diseases. Clinical and genetic heterogeneity make mitochondria-related rare disorders an ideal disease group for an untargeted sequencing approach in genetic diagnosis. Whole exome sequencing (WES) approaches have significantly improved diagnostic yield and improved the understanding of mitochondrial biology, opening potential therapeutic avenues [3]. Autosomal recessive aminoacyl-tRNA synthetases (ARS) deficiencies represent a group of rare inherited MDs with a severe course and multiple organ involvement. Mitochondrial aARS2 enzymes catalyze the addition of amino acids to the newly synthesized polypeptide chain and play an important role in translation. They are the key enzymes responsible for loading amino acids of the same origin into tRNA, and this enzymatic pathway is the most important step in the conversion of genetic information into proteins of interest [4]. The nuclear genome contains 37 ARS genes: 17 in the cytoplasm (known as aARS1), 17 in the mitochondria (known as ARS2), and three encoding enzymes that act in both cellular compartments [4,5]. The first initial letter is the single-letter symbol of the corresponding amino acid, and *DARS2*, *EARS2*, *WARS2*, *SARS2*, and *AARS2* are rare MDs that are the prototypes of this group of mitochondrial aspartate t-RNA synthetase, glutamate t-RNA synthetase, tryptophan t-RNA synthetase, serine t-RNA synthetase, and alanine-tRNA synthetase mutations, respectively. The severe course and clinical heterogeneity result from defects in mt-aARS2, which prevent the synthesis and function of ubiquitous enzymes involved in mitochondrial aminoacylation and translation [5,6]. Reduced or disrupted translation may cause impaired protein folding, and intracellular stress can increase the release of mitokines such as FGF21. It is mainly secreted by the liver, but it may also be expressed in skeletal muscle.

Comprehensive investigations conducted over the past few decades have demonstrated the critical role that FGF21 plays in controlling a wide range of essential metabolic pathways, such as the transmission of insulin signals, the development of muscles, and the reaction to stress in the muscles. FGF21 is believed to help cells and tissues recover their equilibrium after adapting to a stressful environment by coordinating these metabolic pathways [7]. It has been suggested that measuring the concentration of FGF21 in serum could be applied as a first-line diagnostic test to reduce the need for a muscle biopsy in the diagnosis of primary respiratory chain deficiencies affecting muscles in adults and children [8]. In our study, we aimed to evaluate serum FGF21 levels, a mitokine that has recently come to the forefront as an indicator of mitochondrial stress in patients with mt-aARS deficiency.

## 2. Results

### 2.1. Study Groups

The demographic characteristics of six affected patients from six unrelated families and their molecular findings are given in Table 1A,B.

#### 2.1.1. Clinical Findings

The clinical symptoms of the patients were highly heterogeneous. The patient with a pathogenic variant in the *SARS2* gene presented with renal involvement, whereas developmental retardation and lactate elevation were predominant in patients with pathogenic variants in *AARS2* and *WARS2* genes. Headache and cranial involvement were the main complaints in patients with the *DARS2* pathogenic variant. Mental/developmental retardation and severe cerebellar atrophy were present in the patient with a pathogenic variant in the *EARS2* gene.

#### 2.1.2. Laboratory Findings

The means, standard deviations, medians, and interquartile ranges for each of the continuous variables in participants were shown in Table 2. No statistically significant difference was observed between the mean age and gender distribution of the control and patient groups (*p* = 0.743, *p* = 0.732). Also, no statistically significant difference was observed between the mean Weight (kg), Height (cm), and BMI (*p* = 0.682, *p* = 0.815, *p* = 0.896, respectively). Lactate (mg/dL), Pyruvate (mg/dL), and FGF21 (pg/mL) levels in the patient group were significantly higher compared with the control group (*p* = 0.001, *p* = 0.015, *p* = 0.001, respectively).

The area under the Receiver Operating Characteristic (ROC) curve was calculated to determine the efficiency of Lactate (mg/dL), Pyruvate (mg/dL), and FGF21 (pg/mL) variables in the differential diagnosis of mitochondrial aminoacyl-tRNA synthetase deficiency (Figure 1). The area under the ROC Curve values for Lactate (mg/dL), Pyruvate (mg/dL), and FGF21 (pg/mL) in the differential diagnosis of mitochondrial aminoacyl-tRNA synthetase deficiency are presented in Table 3A. Sensitivity, Specificity, Positive Predicted Value, and Negative Predicted Value values for all three markers are presented in Table 3B.

After calculating the means and standard deviations of the case and control groups, two patients were named Case Group 2 (SARS2 and WARS2) because their FGF21 levels were much higher than those of the others. The other four patients were designated Case Group 1. Case Group 2 had higher FGF21 levels than Case Group 1. A violin plot was used to visualize the difference in FGF21 levels between groups, and since the difference in FGF21 levels between individuals was very high, a logarithmic transformation was applied (Figure 2).

## 3. Discussion

In this study, we analyzed FGF21 levels in the serum of patients with MDs to assess whether it is a feasible biomarker for human mitochondria-related disorders. Serum FGF21 levels in patients with molecularly confirmed mitochondrial disease were significantly higher than in the control group, consistent with the literature. However, the small sample size is a major limitation due to the rarity of the disease. Although the finding of 100% sensitivity and specificity is remarkable, it should be kept in mind that this may be a result of the small and specific cohort.

In the last decade, serum FGF21 levels have been proposed as a biomarker for mitochondrial diseases, and many studies have shown increased levels [9,10,11,12,13,14,15]. In a study involving a total of 140 participants (54 patients with mitochondrial disease, 20 adult patients with neuromuscular disease, and 66 healthy volunteers) to determine the validity and reliability of FGF21 as a biomarker, the serum levels were found to be more sensitive than the serum creatine kinase, lactate, and pyruvate, which are classical biomarkers [9]. In another study, a total of 13 biomarkers, including lactate, pyruvate, creatine kinase (CK), amino acid profiles, glutathione, malondialdehyde, GDF-15, FGF21, gelsolin, neurofilament light chain, and circulating cell-free mtDNA, were comprehensively examined in MDs. GDF-15, followed by FGF21, has been shown to have the greatest value, although it is not perfect [15]. In our study, only two patients had increased lactate and pyruvate levels. FGF21 was high compared to the control in all patients. This finding supported the fact that FGF21 was more sensitive. Subsequent studies have shown that serum FGF21 is a specific biomarker for caste defects of mitochondrial translation, including mitochondrial transfer RNA mutations and primary and secondary mtDNA deletions. However, normal serum FGF21 does not rule out structural respiratory chain complex or assembly factor defects, which are important to consider in diagnosis [16]. In another study, patients carrying mutations in nuclear DNA had higher FGF21 levels than patients with mtDNA mutations [17].

Aminoacyl-tRNA synthetases are critical enzymes responsible for attaching amino acids to their corresponding tRNA molecules, a pivotal step in translating genetic information into proteins. Each amino acid is paired with a distinct aminoacyl-tRNA synthetase. Mutations affecting tRNA synthetases alter the transport rate of the related amino acids, leading to clinical manifestations dependent on the levels and activity of the associated proteins. The extent of mitochondrial damage and clinical symptoms varies according to the synthesis of proteins linked to the affected amino acid [5,6]. MDs are caused by different molecular mechanisms, and diagnostic cut-off values of FGF21 levels may vary. The strength of our study is that all included patients had similar molecular etiology. The limitation of our study is the small number of cases, but it is normal for rare diseases.

Biomarkers play a pivotal role in indicating the presence or absence of diseases, as well as monitoring their progression and treatment responses. Identifying a single dependable biomarker encompassing all MDs, particularly those involving primary mitochondrial respiratory chain dysfunction, remains exceedingly challenging. Instead, clinicians often rely on combinations of biomarkers, referred to as “biosignatures,” rather than a solitary indicator for diagnostic purposes. While serum biomarkers like lactate and pyruvate find widespread use in diagnostics, they lack specificity for mitochondrial disorders. In certain cases, simultaneous assessments of creatine, plasma amino acids, and urinary organic acids can enhance the biosignature’s utility [18].

In a recent study, FGF21 was identified as a potent metabolic regulator. This signaling protein belongs to the fibroblast growth factor family of endocrine substances and is secreted into the circulation. The liver and brown and white adipose tissues are the main organs where FGF21 is expressed [19]. But the liver is the primary source, and its secretion is induced by nutritional and cellular stress signals. Hepatic FGF21 is produced in response to nutritional stressors such as prolonged fasting, a ketogenic diet, amino acid deprivation, or simple sugar consumption [20]. In our study, we did not identify any significant correlation between high FGF21 levels and the body mass index (BMI) or age of our patients.

In our study, FGF21 levels were significantly higher in patients with different aminoacyl t-RNA synthase deficiencies compared with the control group. We also observed significantly higher FGF21 levels in patients with SARS2 and WARS2 variants, suggesting a potential genotype–biomarker correlation that should be investigated in the future. This supports the idea that the degree of mitochondrial damage and clinical findings vary according to the amount of synthesis of related amino acid-related proteins. This study also highlights the importance of this biomarker in the diagnosis of MDs caused by different mt-aARS deficiencies and sheds light on the ongoing evaluation of various approaches in this field.

## 4. Materials and Methods

### 4.1. Study Design and Participants

This study was carried out with six probands who were followed up in the Department of Pediatric Metabolism of the Istanbul University Faculty of Medicine and whose molecular diagnosis was finalized by detecting mutations in the aARS genes. The control group consisted of 12 healthy volunteers. Exome sequencing revealed nine different pathogenic variants in five genes encoding aminoacyl-tRNA synthetase (*AARS2*, *EARS2*, *DARS2*, *SARS2*, and *WARS2*). Sanger sequencing was performed for the segregation of pathogenic alterations. The study protocol was approved by the local ethics committee of our hospital (08.06.2018/11; approved on 8 June 2018). After a detailed explanation of the aims and scope of the study in accordance with the principles of the World Medical Association Declaration of Helsinki, a consent form was obtained from the patient’s parents. Written informed consent to participate in this study was provided by the participants’ legal guardian/next of kin.

### 4.2. Biochemical Analysis

All patients were in a stable phase during the phlebotomy. After fasting for approximately 4 h, the blood sample taken into a gel tube (Becton, Dickinson and Company (BD), Franklin Lakes, NJ, USA) was left for 45 min and centrifuged at 4000 rpm for 10 min. The supernatant was stored at −80 °C until the time of analysis. The analysis was performed by enzyme-linked immunosorbent assay (ELISA) after all samples were collected. Serum FGF21 measurements (Catalogue No. DF2100, Quantikine ELISA Kit, R&D Systems, Inc., Minneapolis, MN, USA) were performed using the sandwich enzyme immunoassay technique, and results were evaluated in pg/mL. The assay is based on the method of quantitative sandwich enzyme immunoassay, and the intra-assay and inter-assay coefficients of variation (CV%) were 2.9–3.9% and 5.2–10.9%, respectively. The traditional biomarkers lactate (LA) and pyruvate (PA) were evaluated in plasma together with FGF21. Plasma lactate levels were analyzed using a colorimetric method in the ADVIA Chemistry LAC original reagent in the ADVIA Chemistry XPT autoanalyzer (Siemens Healthcare Diagnostics, Forchheim, Germany). Plasma pyruvate levels were measured spectrophotometrically with a commercial kit according to the manufacturer’s instructions (Ben S.r.l. Biochemical Enterprise, Milan, Italy). Pyruvate and lactate results were expressed as mg/dl. Reference ranges for lactate and pyruvate are 4.5–20 mg/dL and 0.3–0.9 mg/dL, respectively.

### 4.3. Statistical Analysis

This study performed statistical analyses using the NCSS (Number Cruncher Statistical System) 2007 Statistical Software (Version 07, Kaysville, UT, USA) package program. In addition to descriptive statistical methods (mean, standard deviation, median, interquartile range), the Shapiro–Wilk normality test was used to evaluate the distribution of variables, the independent *t*-test was used to compare paired groups of variables with normal distribution, Mann–Whitney U test was used to compare paired groups of variables that did not show normal distribution, Fisher’s exact test was used to compare qualitative data. The results were evaluated at a significance level of *p* < 0.05.

## 5. Conclusions

Serum FGF21 levels outperform other classical biomarkers as a sensitive indicator of mitochondrial disease and provide a valuable diagnostic tool to complement or support genetic investigations. However, we show that serum FGF21 is clinically useful in patients with genetically confirmed mitochondrial aminoacyl-tRNA synthetase deficiency, thereby enabling a more rigorous evaluation of FGF21 as a biomarker with specific cut-off values across different mitochondrial disease subgroups. Assessment of FGF21 levels may provide insight into the level of mitochondrial damage. Investigation of the biochemical mechanisms underlying the different levels of damage caused by different aminoacyl tRNA synthetases will be important in terms of elucidating clinical heterogeneity. This study also sheds light on the use of FGF21 as a biomarker for the identification of secondary mitochondrial damage in chronic diseases.

## Figures and Tables

**Figure 1 ijms-26-09525-f001:**
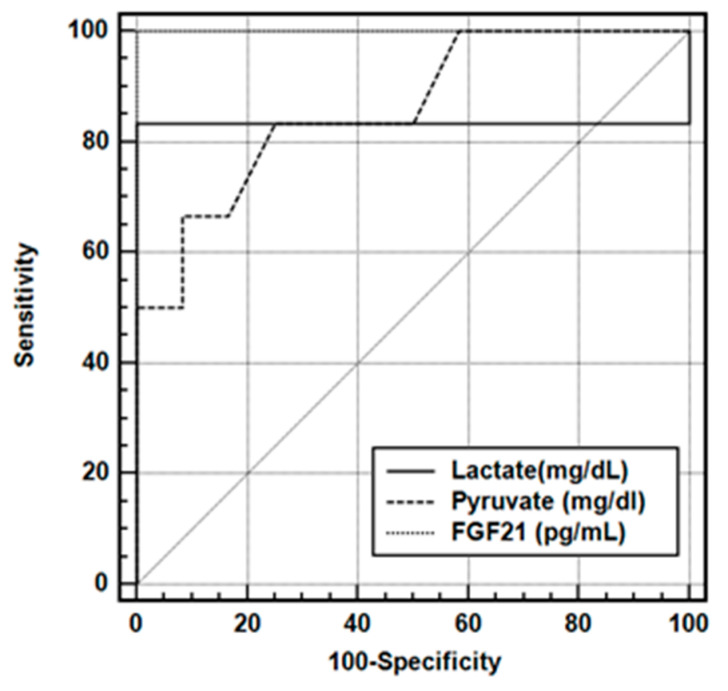
Area under the curve for biomarkers in the differential diagnosis of mitochondrial aminoacyl-tRNA synthetase deficiency.

**Figure 2 ijms-26-09525-f002:**
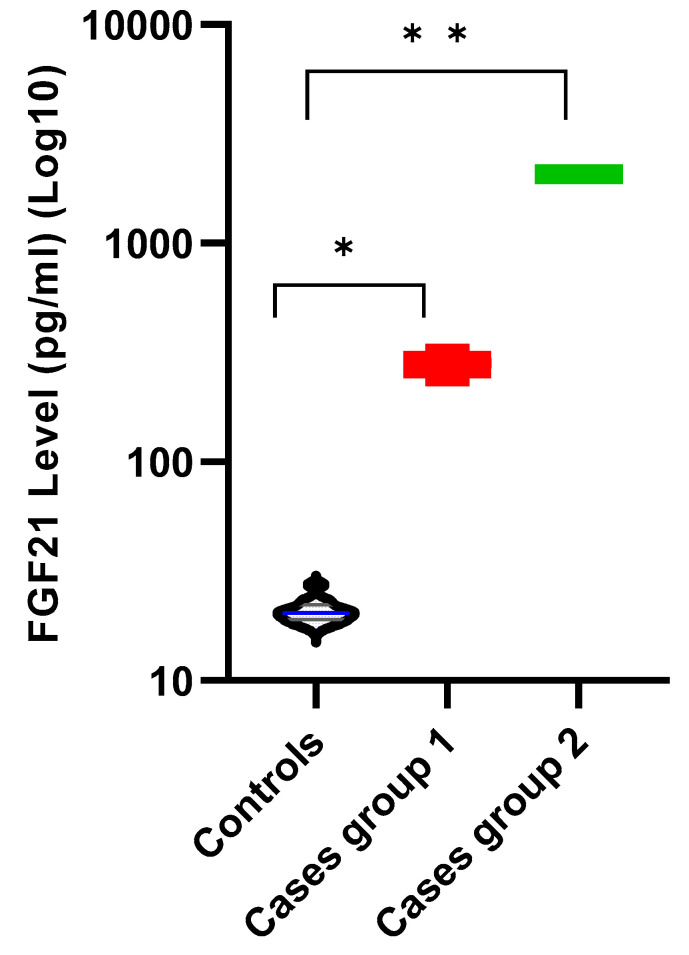
FGF21 levels of case groups and control groups. * Case Group 1: *AARS2* (n = 2), *EARS2*, and *DARS2*. ** Case Group 2: *SARS2* and *WARS2*.

**Table 1 ijms-26-09525-t001:** (**A**) The demographic characteristics and biochemical findings of patients. (**B**) The genetic characteristics and molecular findings of patients.

(A)								
Cases	Gender	Age (Years)	Weight (kg)	Height (cm)	BMI kg/m^2^	Lactate (mg/dL)	Pyruvate (mg/dL)	FGF21 (pg/mL)
C1	M	3.7	9.18	81.5	13.8	49.9	0.59	3495
C2	M	8.1	20.1	120.5	13.8	45.7	0.75	26,661
C3	F	3.4	13.7	96	15.1	8.7	0.37	298.44
C4	M	15.3	46.5	162	17.7	19.8	0.45	233.8
C5	F	6.5	19.5	111.5	15.7	19.5	0.42	2029.45
C6	M	2.9	9.8	88	12.6	26	0.47	2117.14
**(B)**								
**Cases**	**Parental** **Consanguinity**	**Zygosity**	**Gene**	**Variant in Nucleotide**	**Variant in Peptide**			
C1	Yes	Hom	*AARS2*	c.302G>T	p.(Arg101His)			
C2	No	Com.het	*AARS2*	c.277C>T	p.(Arg93Ter)			
C3	No	Com.het	*EARS2*	c.845C>G	p.(Ser282Cys)			
C4	No	Com.het	*DARS2*	c.319C>T	p.(Arg107Cys)			
C5	Yes	Hom	*SARS2*	c.1283delC	p.(Pro428Leufs*3)			
C6	Yes	Hom	*WARS2*	c.492+2T>C	p.(?)			

(**A**) M, male; F, female; BMI, body mass index. Reference ranges of Lactate and Pyruvate are 4.5–20 mg/dL and 0.3–0.9 mg/dL, respectively. (**B**) Hom., homozygous; Com.het., compound heterozygous. The “?” symbol is used because the protein effects of the variants causing the splicing error are unknown.

**Table 2 ijms-26-09525-t002:** Comparison of demographic characteristics and laboratory findings of patient and control groups.

		Control Group		Patient Group		*p*
Age (years)	Mean ± SD	6.73 ± 4.01		6.65 ± 4.70		0.743 ^†^
	Median (IQR)	5.6 (3.55–8.13)		5.1 (3.28–9.9)		
Gender	Male (n, %)	7,	58.33%	4,	66.67%	0.732 +
	Female (n, %)	5,	41.67%	2,	33.33%	
Weight (kg)	Mean ± SD	20.33 ± 13.09		19.8 ± 13.88		0.682 ^†^
	Median (IQR)	16.55 (10.73–20.8)		16.6 (9.65–26.7)		
Height (cm)	Mean ± SD	111.69 ± 27.31		109.92 ± 29.36		0.815 ^†^
	Median (IQR)	104.8 (90.5–124.73)		103.75 (86.38–130.88)		
BMI (kg/m^2^)	Mean ± SD	14.91 ± 2.03		14.78 ± 1.80		0.896 *
Lactate (mg/dL)	Mean ± SD	5.68 ± 0.60		28.27 ± 16.18		**0.001 ^†^**
	Median (IQR)	5.75 (5.13–6.18)		22.9 (16.8–46.75)		
Pyruvate (mg/dL)	Mean ± SD	0.38 ± 0.05		0.51 ± 0.14		**0.015 ^†^**
	Median (IQR)	0.375 (0.33–0.42)		0.46 (0.41–0.63)		
FGF21 (pg/mL)	Mean ± SD	20.89 ± 2.63		882.49 ± 923.60		**0.001 ^†^**
	Median (IQR)	20.33 (19.01–22.24)		323.97 (258.41–2051.37)		

SD, standard deviation; IQR, interquartile range. * Independent *t* test; ^†^ Mann–Whitney U test; + Fisher’s exact test.

**Table 3 ijms-26-09525-t003:** Differential diagnosis of mitochondrial aminoacyl-tRNA synthetase deficiency. AUC for biomarkers (**A**) and laboratory performance assessments (**B**).

(A)							
		AUC		SE		95% CI	
Lactate (mg/dL)		0.933		0.114		0.786–0.962	
Pyruvate (mg/dL)		0.861		0.105		0.618–0.974	
FGF-21 (pg/mL)		1.000		0.000		0.813–1.000	
**(B)**							
	**Criterion**	**Sensitivity**	**Specificity**	**PPV**	**NPV**	**LR (+)**	**LR (−)**
Lactate (mg/dL)	>6.7 *	83.33	100.00	100.0	92.3	N/A	0.17
Pyruvate (mg/dL)	>0.41 *	83.33	75.00	62.5	90.0	3.33	0.22
FGF21 (pg/mL)	>27.4 *	100.00	100.00	100.0	100.0	N/A	0.00

AUC, area under the curve; SE, standard error; PPV, positive predictive value; NPV, negative predictive value; LR (+), positive likelihood ratio; LR (−), negative likelihood ratio; N/A, not applicable. * cut-off value

## Data Availability

The datasets presented in this study can be found in online repositories. The names of the repository/repositories and accession number(s) can be found at the following (accessed 15 July 2025): https://www.ncbi.nlm.nih.gov/snp/, SCV001423134.1; https://www.ncbi.nlm.nih.gov/snp/, SCV001423135.1; https://www.ncbi.nlm.nih.gov/snp/, SCV001571651.1; https://www.ncbi.nlm.nih.gov/snp/, SCV001571652.1.

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
