# Peer review of "Evaluation of Serum FGF21 Levels in Patients with Mitochondrial Aminoacyl-tRNA Synthetase Deficiency"

_ijms, 2025, doi:10.3390/ijms26199525_

Round 1

Reviewer 1 Report

Comments and Suggestions for Authors

This manuscript investigates serum Fibroblast Growth Factor 21 (FGF21) as a potential biomarker for mitochondrial aminoacyl-tRNA synthetase (mt-aARS) deficiency. The authors compared serum FGF21, lactate, and pyruvate levels in six molecularly confirmed patients against twelve healthy controls. The study finds that serum FGF21 is significantly elevated in patients and demonstrates superior diagnostic performance compared to traditional biomarkers like lactate and pyruvate within this small cohort. The authors conclude that FGF21 is a highly sensitive and specific biomarker for this condition and suggest that its varying levels may correlate with the specific genetic defect (e.g., SARS2, WARS2 vs. others).

Major Strengths

The study addresses a significant need in the field of rare diseases—mt-aARS deficiency—the identification of reliable, minimally invasive biomarkers. Improving the diagnostic pathway for mitochondrial diseases (MDs) is a valuable contribution. The patient data is presented transparently and in detail. Tables 1a and 1b, which provide individual demographic, clinical, biochemical, and genetic information, are particularly valuable for the rare disease community.

Major Weaknesses and Suggestions for Improvement

- Laboratory Finding: There is a significant inconsistency in the reported Area Under the Curve (AUC) for FGF21. The abstract states the AUC for FGF21 is 0.933. However, Table 3A clearly reports the AUC for FGF21 as 1.000, while the value of 0.933 corresponds to lactate. The perfect right-angle curve for FGF21 in Figure 1 also supports an AUC of 1.000. Please revise.
In addition, the post-hoc division of the six patients into "Cases group 1" (n=4) and "Cases group 2" (n=2) is a major methodological flaw. The rationale for this split is that two patients had much higher FGF21 levels. Performing a statistical comparison between a group of four and a group of two is not statistically robust and is highly prone to type I errors. The conclusion drawn from this comparison is therefore unreliable. The p-value reported in the legend for Figure 2 (**p<0.055) is confusing, non-standard, and not statistically significant at the conventional p<0.05 level. Instead, they can present the violin plot (Figure 2) to visually demonstrate the heterogeneity in FGF21 levels. In the discussion, this observation should be framed as a preliminary finding or a hypothesis that requires validation in a larger cohort. For example, they could state, "We observed markedly higher FGF21 levels in patients with SARS2 and WARS2 variants, suggesting a potential genotype-biomarker correlation that warrants future investigation."

- In Table 3B, the positive likelihood ratio (LR+) and negative likelihood ratio (LR-) for FGF21 are either left blank or marked as 0.00. For a test with 100% sensitivity and 100% specificity, the LR+ is technically undefined (as it involves division by zero specificity), and the LR- is 0. It should be noted appropriately in the table, for instance, by using "Undefined" or "N/A" for LR+ to be precise.

- Discussion: The authors correctly acknowledge the small sample size as a limitation due to the rarity of the disease. While unavoidable, this point should be emphasized more strongly in the discussion and conclusion. The finding of 100% sensitivity and specificity is remarkable but should be interpreted with extreme caution, as it could be an artifact of the small, specific cohort.

Comments on the Quality of English Language

The manuscript is generally well-written. However, a final proofreading could help refine some phrasing for improved clarity and flow.

Author Response

Bu makaleyi incelemek için zaman ayırdığınız için çok teşekkür ederiz. Ayrıntılı yanıtları aşağıda ve ilgili düzeltmeleri/düzeltmeleri yeniden gönderilen dosyalarda vurgulanmış/değişiklik izlerinde bulabilirsiniz.

Yorum: Bu makale, mitokondriyal aminoasil-tRNA sentetaz (mt-aARS) eksikliği için potansiyel bir biyobelirteç olarak serum Fibroblast Büyüme Faktörü 21'i (FGF21) araştırmaktadır. Yazarlar, moleküler olarak doğrulanmış altı hastada serum FGF21, laktat ve pirüvat düzeylerini on iki sağlıklı kontrolle karşılaştırmıştır. Çalışma, serum FGF21'in hastalarda önemli ölçüde yüksek olduğunu ve bu küçük kohortta laktat ve pirüvat gibi geleneksel biyobelirteçlere kıyasla üstün tanı performansı gösterdiğini bulmuştur. Yazarlar, FGF21'in bu durum için oldukça hassas ve spesifik bir biyobelirteç olduğu sonucuna varmış ve değişen düzeylerinin spesifik genetik kusurla (örneğin, SARS2, WARS2 ve diğerleri) ilişkili olabileceğini öne sürmüşlerdir.

Başlıca Güçlü Yönleri

Çalışma, nadir hastalıklar alanında önemli bir ihtiyaca, yani güvenilir ve minimal invaziv biyobelirteçlerin belirlenmesine yöneliktir. Mitokondriyal hastalıklar (MD) için tanı yolunun iyileştirilmesi değerli bir katkıdır. Hasta verileri şeffaf ve ayrıntılı olarak sunulmaktadır. Bireysel demografik, klinik, biyokimyasal ve genetik bilgiler sunan Tablo 1a ve 1b, nadir hastalıklar topluluğu için özellikle değerlidir.

Başlıca Zayıflıklar ve İyileştirme Önerileri

Yorum 1- Laboratuvar Bulgusu: FGF21 için bildirilen Eğri Altındaki Alan (AUC) değerlerinde önemli bir tutarsızlık bulunmaktadır. Özette FGF21 için AUC değerinin 0,933 olduğu belirtilmektedir. Ancak Tablo 3A, FGF21 için AUC değerini açıkça 1,000 olarak bildirirken, 0,933 değeri laktata karşılık gelmektedir. Şekil 1'deki FGF21 için mükemmel dik açılı eğri de 1,000 AUC değerini desteklemektedir. Lütfen gözden geçirin.

In addition, the post-hoc division of the six patients into "Cases group 1" (n=4) and "Cases group 2" (n=2) is a major methodological flaw. The rationale for this split is that two patients had much higher FGF21 levels. Performing a statistical comparison between a group of four and a group of two is not statistically robust and is highly prone to type I errors. The conclusion drawn from this comparison is therefore unreliable. The p-value reported in the legend for Figure 2 (**p<0.055) is confusing, non-standard, and not statistically significant at the conventional p<0.05 level. Instead, they can present the violin plot (Figure 2) to visually demonstrate the heterogeneity in FGF21 levels. In the discussion, this observation should be framed as a preliminary finding or a hypothesis that requires validation in a larger cohort. For example, they could state, "We observed markedly higher FGF21 levels in patients with SARS2 and WARS2 variants, suggesting a potential genotype-biomarker correlation that warrants future investigation."

Response 1: Thank you for your constructive comments. We agreed. The error in the Area Under the Curve (AUC) value reported for FGF21 in the abstract section has been corrected. We indicated the correct expression in the Abstract section, Line 24-25 and colored it yellow; “The area under the ROC Curve for FGF21 in the differential diagnosis of mitochondrial amino-acyl-tRNA Synthetase Deficiency was 1,000      (0,813-1,000).”

Additionally, as suggested, observational data from two patients with significantly higher FGF21 levels were presented only as a violin plot (Figure 2) to visually demonstrate the heterogeneity in FGF21 levels. Lines 34 -39 in the text have been corrected and colored yellow. In the discussion, this observation was framed as a preliminary finding that needs to be confirmed in a larger cohort. "In our results, we observed significantly higher FGF21 levels in patients with SARS2 and WARS2 variants, suggesting a potential genotype-biomarker correlation that should be investigated in the future." The statement was added between lines 200 and 203 and highlighted in yellow.

Comment 2- In Table 3B, the positive likelihood ratio (LR+) and negative likelihood ratio (LR-) for FGF21 are either left blank or marked as 0.00. For a test with 100% sensitivity and 100% specificity, the LR+ is technically undefined (as it involves division by zero specificity), and the LR- is 0. It should be noted appropriately in the table, for instance, by using "Undefined" or "N/A" for LR+ to be precise.

Yanıt 2: Değerli yorumunuz için teşekkür ederiz. Biz de aynı fikirdeyiz. Önerileriniz doğrultusunda, belirtilen düzenleme tamamlandı. Tablo 3B'de, FGF21 için pozitif olasılık oranı (LR+), doğruluğu sağlamak için tabloda uygun şekilde "Uygun Değil" kullanılarak işaretlenmiştir. Tabloda yapılan düzeltmeler sarı renkle vurgulanmıştır.

Yorum 3- Tartışma: Yazarlar, hastalığın nadir görülmesi nedeniyle küçük örneklem boyutunun bir sınırlama olduğunu haklı olarak kabul etmektedir. Kaçınılmaz olsa da, bu nokta tartışma ve sonuç bölümünde daha güçlü bir şekilde vurgulanmalıdır. %100 duyarlılık ve özgüllük bulgusu dikkat çekicidir, ancak küçük ve özgül kohortun bir sonucu olabileceğinden son derece dikkatli yorumlanmalıdır.

Yanıt 3 - Son derece önemli ve faydalı yorumunuz için çok teşekkür ederiz. Biz de aynı fikirdeyiz. "%100 duyarlılık ve özgüllük bulgusu dikkat çekici olsa da, bunun küçük ve özgül kohortun bir sonucu olabileceği akılda tutulmalıdır" yorumu, önerildiği gibi Tartışma bölümünün ilk paragrafına, 148-151. satırlara eklendi (sarı renkle vurgulanmıştır).

Makalemizi inceleyip bize faydalı yorumlarda bulunduğunuz için teşekkür ederiz.

Reviewer 2 Report

Comments and Suggestions for Authors

This manuscript indicated that serum FGF21 levels can serve as an indicator in mt-aARS deficiency patients. The conclusion is convincing. However, several concerns remain.

1, In Table 1a, what are the characteristics and biochemical information of the control persons?

2, In Table 3A and 3B, the abbreviations need to list their full names.

3, Are the serum FGF21 levels elevated and can it be used as an indicator in other mitochondrial deficiency diseases?

4, The capitalization of table numbers needs to be standardized. For example, in Line 88, “Table 1. a.” was displayed, while in Line 127, “3A” was displayed.

Author Response

This manuscript indicates that serum FGF21 levels can serve as an indicator in mt-aARS deficiency patients. The conclusion is convincing. However, several concerns remain.

Comment-1: In Table 1A, what are the characteristics and biochemical information of the control persons?

Response-1: Response 3 - Thank you for your valuable comment. Since the study group is a rare disease, openly sharing biochemical data may be important for other researchers. The control group data was not shared in a tabular form, but it was stated that the data is available to those who request it. The control group data is shown in the table below:

Cont

Gender

Age(years)

Weight (kg)

Height (cm)

BMI (kg/m2)

Lactate (mg/dL)

Pyruvate (mg/dl)

FGF21 (pg/mL)

K1

M

4,1

10,1

82,3

14,9

5,9

0,31

18,961

K2

M

7,9

19,3

122,4

12,8

5,7

0,42

20,469

K3

F

3,7

14,2

99

14,4

4,9

0,33

22,496

K4

M

14,9

47,8

162

18,2

6,1

0,41

20,114

K5

F

6,7

20,5

115

15,5

4,8

0,36

27,359

K6

M

3,1

10,4

92

12,2

5,8

0,34

20,861

K7

F

4,5

11,7

85

16,9

6,2

0,32

20,185

K8

M

8,2

20,9

125,5

13,2

5,1

0,43

17,651

K9

M

3,5

14,1

95

15,6

6,7

0,41

18,446

K10

F

13,8

46,1

161,5

17,6

6,3

0,38

21,472

K11

M

7,1

18,9

110,6

15,4

5,4

0,46

23,457

K12

F

3,3

9,9

90

12,2

5,2

0,37

19,154

Comment-2: 2, In Table 3A and 3B, the abbreviations need to list their full names.

Response-2: We agreed, all abbreviations mentioned in table 3A and table 3B are explained below the table.

*AUC; area under the curve, SE; standard error, PPV; positive predictive value, NPV; negative predictive value, LR (+); positive likelihood ratio, LR (-); negative likelihood ratio, N/A; not applicable

Comment-3: Are the serum FGF21 levels elevated and can it be used as an indicator in other mitochondrial deficiency diseases?.

Response-3: Thanks for your instructive comment. Previous studies and our findings show that circulating serum FGF-21 is elevated in patients with mitochondrial diseases. Further validation of serum FGF-21 as a biomarker warrants long-term follow-up of affected patients, in addition to its measurement during acute episodes of clinical deterioration. Serum FGF-21 levels outperform other classical biomarkers as a sensitive indicator of mitochondrial disease and provide a valuable diagnostic tool to complement or support genetic investigations. However, we show that serum FGF-21 is clinically useful in patients with genetically confirmed mitochondrial tRNA synthetase deficiency, thereby enabling a more rigorous evaluation of FGF-21 as a biomarker with specific cut-off values across different mitochondrial disease subgroups. These statements were added to the concluding paragraph in line with your suggestion (Line 248-253 and highlighted in yellow)

Comment-4: 4, The capitalization of table numbers needs to be standardized. For example, in Line 88, “Table 1. a.” was displayed, while in Line 127, “3A” was displayed.

Response-4We agreed. We corrected and standardized all of them in capital letters, both above the tables and in the text.

We thank you for reviewing our manuscript and giving us useful comments.

Round 2

Reviewer 1 Report

Comments and Suggestions for Authors

Nothing to add

Comments on the Quality of English Language

Consider improving the English language before publication

Author Response

Comment 1-Consider improving the English language before publication.

Response 1- Thank you for your valuable comment. We have reviewed and corrected our English spelling errors. We have submitted the revised manuscript.